# Association of Hand Grip Strength with Mild Cognitive Impairment in Middle-Aged and Older People in Guangzhou Biobank Cohort Study

**DOI:** 10.3390/ijerph19116464

**Published:** 2022-05-26

**Authors:** Ya-Li Jin, Lin Xu, Chao-Qiang Jiang, Wei-Sen Zhang, Jing Pan, Feng Zhu, Tong Zhu, Graham Neil Thomas, Tai-Hing Lam

**Affiliations:** 1Guangzhou Twelfth People’s Hospital, Guangzhou 510620, China; jinyali22@163.com (Y.-L.J.); zwsgzcn@163.com (W.-S.Z.); jcdaise@163.com (J.P.); chifengzhu@hotmail.com (F.Z.); zhutong33@aliyun.com (T.Z.); 2School of Public Health, Sun Yat-sen University, Guangzhou 510080, China; xulin27@mail.sysu.edu.cn; 3Institute of Applied Health Research, University of Birmingham, Birmingham B15 2TT, UK; g.n.thomas@bham.ac.uk; 4School of Public Health, The University of Hong Kong, Hong Kong; hrmrlth@hku.hk

**Keywords:** hand grip strength, mild cognitive impairment, Delayed Word Recall Test

## Abstract

Background: Lower hand grip strength has been linked to cognitive impairment, but studies in older Chinese are limited. We examined the association of hand grip strength with cognitive function in a large sample of older Chinese. Methods: 6806 participants aged 50+ years from the Guangzhou Biobank Cohort Study (GBCS) were included. Relative grip strength was calculated by absolute handgrip strength divided by the body mass index (BMI). Cognitive function was assessed using the Delayed Word Recall Test (DWRT, from 0 to 10) and the Mini Mental State Examination (MMSE, from 0 to 30), with higher scores indicating better cognition. Results: After adjusting for multiple potential confounders, lower absolute grip strength and relative grip strength were significantly associated with lower DWRT (all *p* < 0.05) in all participants. No significant interaction effects between sex and handgrip strength on cognitive impairment were found (*p* from 0.27 to 0.87). No significant association between handgrip strength and total MMSE scores was found in the total sample or by sex (*p* from 0.06 to 0.50). Regarding the individual components of MMSE, lower absolute and relative grip strength were significantly associated with lower scores of the recall memory performance in all participants (*p* from 0.003 to 0.04). Conclusion: We have shown for the first time a positive association of grip strength with recall memory performance, but not general cognitive function in older people, which warrants further investigation.

## 1. Introduction

Mild cognitive impairment (MCI) is a transitional phase between healthy cognitive ageing and dementia, characterized by memory loss and cognitive decline that is not severe enough to have impacts on activities of daily living [1]. Individuals with the “symptomatic pre-dementia stage” have a greater risk of progression to Alzheimer’s disease (AD) [2]. However, there are no effective pharmacologic treatments to slow or cure MCI [3]. Nonetheless, lifestyle modifications including diet, aerobic exercise, mental activity, and social engagement may have small beneficial effects on preventing cognitive decline [4]. Aerobic exercise improves physical function, and the latter is associated with improved activities of daily living, quality of life, and frailty [5]. Measures of physical function, such as hand grip strength, have also been associated with cognitive impairment, mostly assessed using the Mini-Mental State Examination (MMSE) in older people [6,7]. Hand grip strength is a reliable, simple, quick, and inexpensive measure of muscular strength [8]. Absolute handgrip strength represents the lower arm, leg, and core muscle strengths in the standing position, which are closely related to body size [5,9]. Relative handgrip strength (i.e., absolute handgrip strength divided by body mass index (BMI)) has been used in epidemiologic studies to adjust for body size [10]. Previous studies have shown that relative grip strength was associated with cardiovascular risk factors and metabolic diseases [10,11].

However, most of the previous studies describing the positive associations between muscle strength and cognitive functions focused on dementia patients and young people. Our search of PubMed, EMBASE, and CNKI (China Academic Journal Network Publishing Database) using keywords of “handgrip strength” and (“cognitive impairment” OR “cognitive function” OR cognition) up to 31 July 2021 found no study describing the association between relative handgrip strength and cognitive function, or specific domains of cognition such as verbal memory and other general cognitive functions such as global cognition, attention, and executive function. Therefore, the aim of this study was to assess the association of absolute and relative handgrip strength with cognitive function in a large community-based population of the Guangzhou Biobank Cohort Study (GBCS) [12] using baseline cross-sectional data.

## 2. Methods

### 2.1. Participants

GBCS is a three-way collaboration among the Guangzhou Twelfth People’s Hospital and the Universities of Hong Kong and Birmingham. Details of the GBCS have been reported previously [12]. All participants were recruited from the Guangzhou Health and Happiness Association for the Respectable Elders (GHHARE), a large social and welfare organization. The GHHARE included about 7% of residents in this age group, with branches over all districts of Guangzhou. All participants were relatively healthy Guangzhou residents aged 50 or above and without history of stroke, myocardial infarction, pulmonary heart disease, and malignant tumor. The study was approved by the Guangzhou Medical Ethics Committee of the Chinese Medical Association. All participants provided written informed consent before participation.

This study used the third phase of baseline data from GBCS from September 2006 to September 2007. The Delayed Word Recall Test (DWRT) was administered to all baseline participants of GBCS, and MMSE was administered in the third phase of the GBCS. Face-to-face interviews were conducted by trained nurses to collect information on demographic characteristics, lifestyle, and personal and family medical history. Fasting blood samples were drawn from all subjects after an overnight fast. Lipids, glucose, markers of liver, renal, and cardiac function, rheumatoid factor, and high sensitivity C-reactive protein were assayed in the hospital laboratory. Weight, standing height, sitting height, waist circumference, and hip circumference were measured with light indoor clothing and without shoes. Physical examination included measurement of blood pressure, anthropometric indices, 12-lead electrocardiography, pulmonary function testing, and chest radiograph. [12]. Body mass index (BMI) was calculated using measured weight and height as kilograms divided by meters squared.

### 2.2. Cognitive Measures

The Delayed Word Recall Test (DWRT) [13] and Mini-Mental State Examination (MMSE) [14] were used to assess cognitive function. DWRT is a test of verbal learning and recent memory requiring recall of a word list. During the interview, 10 simple Chinese words (soy sauce, arm, letter, chairman, ticket, grass, corner, stone, book, and stick) were read out one by one to participants, who were asked to recall the words. This procedure was repeated three times, and then after 5 min, participants were asked to freely recall as many words as possible. Participants were given a score of 1 for each correct word, with a maximum score of 10. The scores from this fourth repetition were analyzed as DWRT score. Cognitive impairment was defined as DWRT score < 4, corresponding to 1 standard deviation (SD) below the mean (M = 5.9; SD = 2.0). Participants were classified into 3 groups (<4 as cognitive impairment, 4–6 as median cognitive function, ≥7 as better cognitive function) according to DWRT scores [14,15]. MMSE was added at the second examination to assess cognitive function. The 30 points on the 11 MMSE items test measures 5 cognitive components including orientation (0–10 scores), memory (0–3 scores), attention and calculation (0–5 scores), recall memory (0–3 scores), and language (0–9 scores), with total scores ranging from 0 to 30. Poor cognitive function was defined by an MMSE score < 25 corresponding to 1 standard deviation (SD) below the mean (M = 27.7; SD = 2.4) [15,16]. The cut-off scores were not exactly one standard deviation below the means for DWRT or MMSE but were rounded to the nearest whole number. The MMSE values were modelled as both categorical and continuous variables.

### 2.3. Hand Grip Strength Measurements

Grip strength was assessed using a Jamar Hydraulic Hand Dynamometer in a standing position. The dynamometer was explained and demonstrated to participants before use. Participants were asked to have a practice session before testing hand grip strength. Participants were randomly assigned to start the test with their dominant or non-dominant hand. To follow the pre-specified standard operation protocol, the grip strength of each hand was tested two times, and the average value of the repeated measurements was calculated, expressed as kilograms. The maximal reading of the average grip strength in right and left hands was used as the absolute grip strength (AGS). Maximum relative grip strength (RGS_max_) was calculated by AGS divided by BMI. Average relative grip strength (RGS_mean_) was calculated by the average grip strength in both hands divided by BMI. Relative grip strength in the left- or right-hand (RGS_left_/RGS_right_) was calculated by the average grip strength in the left- or right-hand divided by BMI [17,18].

DWRT has been found to be an efficient instrument for discriminating normal cognition and those with mild dementia, with a correct classification value up to 95.2% [19]. The 6-month test–retest reliability in 26 subjects with normal cognition was found to be 0.75, which was acceptable [19]. The DWRT was designed specifically to be used in population-based epidemiological studies or in screening examinations [20,21]. The reliability and validity assessment of the Chinese version of MMSE showed that MMSE was in general a reliable screening test for cognitive impairment [22,23,24]. The sensitivity of diagnosis of dementia or mild cognitive impairment in the general population by the MMSE was about 65% to 85% according to previous studies [25,26,27,28].

### 2.4. Statistical Analysis

All data analyses were performed using IBM SPSS Statistics for Windows (version 26.0, Armonk, NY, USA). Pearson χ^2^ test and one-way analysis of variance (ANOVA) were used to compare categorical and continuous variables between different DWRT groups, respectively. Continuous variables were summarized as mean and standard deviation (SD), and ordinal variables as numbers and percentages. Hand grip strength was compared between different DWRT groups using analysis of covariance (ANCOVA). Multivariable linear regression was used to analyze the association of hand grip strength with cognitive function, giving adjusted regression coefficients (β) and 95% confidence intervals (CIs). Potential confounders adjusted were sex, age, education, smoking status, physical activity, fasting glucose, and systolic blood pressure. Because of the differences in hand grip strength between men and women, a sex interaction on the association between hand grip strength and cognitive impairment was tested, and we found no significant interactions (*p* from 0.27 to 0.87).

## 3. Results

The participants were aged 50 to 96 years, with the mean (SD) age being 63.7 (7.6) and 59.6 (7.6) years for men and women, respectively. The mean DWRT score was 5.9 (SD = 2.0), and the median MMSE score was 28 (interquartile range = 2) for all participants. In total, 6806 participants were included in the present study, and of these, 148 did not have MMSE data. Table 1 shows that in both men and women, compared to those with a higher DWRT score, those with a DWRT score of <4 had older age, lower education, inactive physical activity, and higher systolic blood pressure (all *p* < 0.001). In women, those with lower DWRT score had higher proportions of current smokers and manual workers, and higher levels of fasting glucose and waist circumference (all *p* < 0.001).

Table 2 shows that after adjusting for sex, age, education, smoking status, physical activity, fasting glucose, and systolic blood pressure, overall MMSE scores and all the sub-domains were significantly associated with DWRT in all participants (*p* < 0.001 for orientation, attention and calculation, recall memory, and language (*p* = 0.04 for memory). The adjusted β (95% CI) was 0.07 (0.06, 0.07) for total MMSE scores, 0.12 (0.10, 0.14) for orientation, 0.11 (0.002, 0.22) for memory, 0.08 (0.07, 0.09) for attention and calculation, 0.13 (0.11, 0.14) for recall memory, and 0.11 (0.08, 0.13) for language.

Table 3 shows in all participants that after adjusting for potential confounders, both lower absolute grip strength and relative grip strength were significantly associated with lower DWRT in all participants (all *p* < 0.05). Linear trends were also significant. Adjusted β (95% CI) was 0.09 (0.04, 0.14, or 0.15) for all four relative grip strength measures and 0.004 (0.002, 0.007) for absolute grip strength. The associations were similar and remained significant in men and in women (all *p* < 0.05). Table 4 shows that after similar adjustment, no significant association or linear trend between hand grip strength and MMSE was found in the total group, or in men and in women specifically.

Table 5 shows that both absolute and relative grip strength were significantly associated only with the recall memory components of MMSE in all participants (*p* from 0.003 to 0.04) except RGS_right_ (*p* = 0.14). The adjusted β (95% CI) for recall memory was 0.07 (0.01, 0.15) for RGS_max_, 0.08 (0.003, 0.15) for RGS_mean_, 0.09 (0.02, 0.17) for RGS_left_, and 0.005 (0.002, 0.009) for AGS.

## 4. Discussion

To our knowledge, this is the first study showing that hand grip strength was associated with recall memory performance using two separate but complementary methods. Although no association between grip strength and total MMSE score was found, poorer grip strength was significantly associated with lower scores of DWRT and the recall memory domain of the MMSE, indicating that low grip strength might be more closely and significantly associated with the memory recall aspects of cognition in older people.

A positive association of hand grip strength and cognitive function has been reported previously [29,30,31,32]. For example, a cross-sectional study in the U.S. [29] showed that handgrip strength was a simple risk-stratifying method for identifying populations at risk for poorer cognitive function assessed by MMSE. In a community-based cross-sectional study in Japan [30] of older adults, poorer hand grip strength was associated with lower global cognitive function measured by the Montreal Cognitive Assessment. Moreover, in Asia, a cross-sectional study from the Korean Longitudinal Study of Aging showed that handgrip strength was associated with a higher risk of MCI assessed by MMSE in older Koreans [31]. Similarly, a cross-sectional study of older adults in China [32] found that hand grip strength was associated with a higher risk of MCI assessed by MMSE. There were also some longitudinal studies [33,34,35] examining the association between handgrip strength and cognitive impairment. A prospective cohort study [33] found that reduced handgrip strength at baseline demonstrated a statistically significant decline in cognitive function assessed by MMSE over a 7-year period in older Mexican Americans. In a prospective population-based study from the Netherlands [34], after 4 years of follow-up, baseline cognitive performance measured by the neuropsychological test battery was associated with a decline in handgrip strength. In another prospective observational study in China [35], weaker handgrip strength significantly correlated with lower MMSE after 4 years. However, there were mixed results on the association between hand grip strength and cognitive performance in other studies [36,37,38]. In the U.S. Women’s Health Initiative Memory Study (WHIMS) [36], the baseline cross-sectional study found that a decrease in hand grip strength in older women was not associated with a decline in cognitive function measured by the MMSE. Another cross-sectional study [37] of French women aged 75 and older also found no significant association between hand grip strength and cognitive impairment measured by the Short-Portable-Mental-State-Questionnaire (SPMSQ). A prospective cohort study of Italians [38] did not find a significant association between baseline handgrip strength and the onset of cognitive impairment assessed by MMSE. The studies above did not examine components of MMSE separately, which might lead to unclear or nonspecific inconsistent results. Our study adds to the literature by highlighting that low grip strength was more closely related to recall memory performance, which is consistent with several studies [7,39]. For example, grip strength was associated with fewer retrospective memory complaints in a cross-sectional study [39] of U.S. young adults (mean age = 20.7 years). Another study [7] using baseline data from the UK Biobank on adults aged 37–73 years found that grip strength was significantly associated with working memory. Our study is the first study to show the association between relative handgrip strength and recall memory in older people.

Hand grip strength, a measure of body function, has been suggested as an indicator of current and future health status. A Danish study [40] of 52 acutely admitted older patients aged ≥65 years found that the inter-rater reliability for absolute grip strength was 0.95. Another study [41] of 76 older people with dementia found that the test–retest reliability of the absolute handgrip strength test in those with borderline, mild, and moderate dementia was 0.98, 0.97, and 0.96, respectively, suggesting the handgrip strength test has excellent reliability. Although based on small samples, these findings support that absolute hand grip strength is a reliable measure of general muscular strength, which is also inexpensive and readily accessible. However, absolute handgrip strength was closely related to body size, i.e., people with greater body size generally have greater grip strength [42]. Analyses adjusting for BMI may partially alleviate this issue [11]. Hence, relative strength of a BMI-standardized measure was used by previous studies mainly for examining associations with cardiovascular health and metabolic diseases [10,18]. A general population cross-sectional study [10] in China found that relative handgrip strength was a better predictor of metabolic profile and metabolic disease than absolute handgrip strength. Likewise, a U.S. civilian non-institutionalized population sample also showed that higher relative handgrip strength was associated with more favorable cardiovascular risk profiles, suggesting that relative grip strength might be a useful measure of muscle strength [11].

Hand grip strength has been adopted as an indicator of overall strength and physical function in older adults [43]. Muscle strength can be improved by physical activity, which might not only help to improve physical function but can also have positive effects on cognition and brain function [44]. Physical activity has been associated with improved learning and memory [45] and can offset age-related cognitive decline reducing MCI [46,47,48] and has been associated with slower decline in working memory in patients with Alzheimer’s disease [49]. The underlying mechanisms may be related to an improvement in cerebral circulation after exercise, including an increase in blood flow and oxygen supply to the brain [50]. Exercise might affect synaptic plasticity and brain function through modulating energy metabolism through brain-derived neurotrophic factor (BDNF) [49,51,52]. Physical activity has been shown to reduce inflammatory responses and oxidative stress markers [53,54] which contribute to both vascular dementia and neurological damage. Notably, inflammatory cytokines may have some adverse effects on muscle mass-promoting cachexia [55]. Additionally, inflammatory cytokines may have effects on skeletal muscle, both directly through reducing muscle protein synthesis and catabolism, and indirectly through tissue dysregulation and the hypothalamus–pituitary–adrenal axis (i.e., reducing food intake and promoting cachexia), which reduce the beneficial effect of physical activity [56]. Grip strength may therefore act as a surrogate with which to monitor the impact of these processes generally and cognition in particular.

A main strength of the present study was the use of two different measures of cognition. The DWRT assesses short-term verbal memory, and the MMSE incorporates five neurological aspects: orientation, memory, attention and calculation, recall, and language [13,14]. The results consistently showed significant associations of hand grip strength with the DWRT and the memory recall component of MMSE, whilst the associations with other aspects of general cognition function in MMSE were not significant. However, there were several limitations of this study. This is a cross-sectional study, and therefore the causal relationship between hand grip strength and cognition could not be established. Additionally, as all participants were older people and those with severe cognitive impairment could not participate in this study, selection bias due to survival cannot be ruled out. Finally, as all data were collected from a single city in China, the generalize applicability to other settings may be limited.

## 5. Conclusions

In conclusion, this is the first study demonstrating a positive association of handgrip strength with recall memory performance, but not general cognitive function in older people. The causality of the association warrants further investigation.

## Figures and Tables

**Table 1 ijerph-19-06464-t001:** Characteristics of the Delayed Word Recall Test (DWRT) groups in 1696 men and 5110 women from the Guangzhou Biobank Cohort Study by sex.

	Delayed Word Recall Test (DWRT) Scores			*p* Value
	≥7	4–6	<4	
Men				
Number (rates%)	569 (33.5)	909 (53.6)	218 (12.9)	
Age (years)	61.6 ± 6.8	63.9 ± 7.4	67.7 ± 8.2	<0.001
Education, %				
Primary or below	16.9	28.4	48.2	<0.001
Middle school	62.7	56.2	45.4	
College or above	20.4	15.4	6.4	
Smoking status, %				
Never	40.4	36.9	32.7	0.08
Former	24.3	28.5	24.9	
Current	35.3	34.7	42.4	
Occupation, %				
Manual	27.0	26.6	31.2	0.12
Non-manual	45.4	50.6	45.0	
Other	27.7	22.8	23.9	
Physical activity, %				
Inactive	53.2	54.9	63.6	0.001
Minimally active	35.3	37.7	31.1	
Active	11.5	7.4	5.3	
Fasting glucose (mmol/L)	5.8 ± 1.7	5.8 ± 1.6	5.9 ± 1.6	0.60
Total cholesterol (mmol/L)	5.6 ± 1.0	5.7 ± 1.6	5.6 ± 1.1	0.22
Systolic blood pressure (mmHg)	129 ± 20	132 ± 21	136 ± 25	<0.001
Diastolic blood pressure (mmHg)	75 ± 11	75 ± 11	75 ± 12	0.97
Waist circumference (cm)	80.6 ± 8.8	80.5 ± 9.1	80.4 ± 9.6	0.95
Body mass index (kg/m^2^)	23.5 ± 3.1	23.5 ± 3.1	23.2 ± 3.3	0.18
Women				
Number (rates, %)	2167 (42.4)	2462 (48.2)	481 (9.4)	
Age (years)	57.8 ± 6.5	60.2 ± 7.8	64.1 ± 8.6	<0.001
Education, %				
Primary or below	26.5	47.8	74.2	<0.001
Middle school	64.3	47.4	24.1	
College or above	9.1	4.8	1.7	
Smoking status, %				
Never	98.1	96.7	94.8	<0.001
Former	0.8	1.7	1.9	
Current	1.2	1.6	3.3	
Occupation, %				
Manual	21.2	26.2	34.7	<0.001
Non-manual	49.2	48.0	43.2	
Other	29.7	25.8	22.1	
Physical activity, %				
Inactive	73.3	63.3	58.6	<0.001
Minimally active	21.6	28.4	34.3	
Active	5.1	8.2	7.1	
Fasting glucose (mmol/L)	5.6 ± 1.4	5.8 ± 1.7	5.9 ± 1.9	<0.001
Total cholesterol (mmol/L)	6.0 ± 1.1	6.1 ± 1.1	6,0 ± 1.1	0.28
Systolic blood pressure (mmHg)	124 ± 21	128 ± 23	131 ± 22	<0.001
Diastolic blood pressure (mmHg)	71 ± 11	72 ± 11	71 ± 11	0.07
Waist circumference (cm)	75.6 ± 8.7	76.9 ± 8.9	78.5 ± 8.7	<0.001
Body mass index (kg/m^2^)	23.8 ± 3.4	24.0 ± 3.4	24.1 ± 3.4	0.07

Results are shown as mean ± SD, except for numbers and percentages.

**Table 2 ijerph-19-06464-t002:** Relationships between Delayed Word Recall Test (DWRT) and Mini Mental State Examination (MMSE) and its components.

		DWRT Scores (0–10)		Adjusted Standardized Beta-Coefficient (95% CI)	*p* for Trend
	≥7	4–6	<4		
MMSE scores (0–30)	28.15 (28.07, 28.24)	27.63 (27.56, 27.71)	25.94 (25.76, 26.11)	0.07 (0.06, 0.07)	<0.001
Orientation (0–10)	9.72 (9.69, 9.75)	9.60 (9.57, 9.63)	9.06 (8.99, 9.13)	0.12 (0.10, 0.14)	<0.001
Memory (0–3)	2.99 (2.98, 2.99)	2.98 (2.98, 2.99)	2.97 (2.96, 2.98)	0.11 (0.002, 0.22)	0.04
Attention and calculation (0–5)	4.41 (4.36, 4.45)	4.22 (4.18, 4.26)	3.58 (3.48, 4.26)	0.08 (0.07, 0.09)	<0.001
Recall memory (0–3)	2.29 (2.26, 2.32)	2.10 (2.08, 2.13)	1.70 (1.64, 1.77)	0.13 (0.11, 0.14)	<0.001
Language (0–9)	8.67 (8.65, 8.70)	8.61 (8.59, 8.63)	8.35 (8.30, 8.41)	0.11 (0.08, 0.13)	<0.001

Results are shown as mean (95% confidence interval), except for numbers, adjusted for age, sex, education, smoking status, physical activity, fasting glucose, systolic blood pressure, and BMI, and R^2^ from 0.31 to 0.48.

**Table 3 ijerph-19-06464-t003:** Multivariable linear regression of hand grip strength on different Delayed Word Recall Test (DWRT) groups for males and females.

		DWRT Scores (0–10)		Adjusted Beta-Coefficient (95% CI)	*p* for Trend
	≥7	4–6	<4		
		Total			
Number of subjects	2736	3371	699		
Relative grip strength _max_ ^a^	1.06 (1.05, 1.07)	1.05 (1.04, 1.06)	1.02 (0.99, 1.04)	0.09 (0.04, 0.14)	0.001
Relative grip strength _mean_ ^a^	1.03 (1.02, 1.04)	1.02 (1.01, 1.03)	0.98 (0.96,1.01)	0.09 (0.04, 0.15)	0.001
Relative grip strength _left_ ^a^	1.04 (1.03, 1.05)	1.03 (1.02, 1.04)	0.99 (0.97, 1.01)	0.09 (0.04, 0.14)	0.001
Relative grip strength _right_ ^a^	1.02 (1.01, 1.03)	1.01 (0.99, 1.02)	0.98 (0.96, 1.00)	0.09 (0.04, 0.14)	0.001
Absolute grip strength, kg ^b^	24.88 (24.65, 25.11)	24.64 (24.44, 24.85)	23.78 (23.32, 24.24)	0.004 (0.002, 0.007)	0.001
		Men			
Number of subjects	569	909	218		
Relative grip strength _max_ ^c^	1.46 (1.43,1.49)	1.45 (1.42, 1.47)	1.39 (1.34, 1.43)	0.09 (0.01, 0.18)	0.03
Relative grip strength _mean_ ^c^	1.42 (1.39, 1.45)	1.41 (1.38, 1.43)	1.35 (1.30, 1.39)	0.10 (0.01, 0.18)	0.03
Relative grip strength _left_ ^c^	1.43 (1.40, 1.46)	1.42 (1.40, 1.44)	1.36 (1.31, 1.41)	0.09 (0.01, 0.18)	0.03
Relative grip strength _right_ ^c^	1.41 (1.38, 1.44)	1.39 (1.37, 1.42)	1.33 (1.29, 1.38)	0.09 (0.01, 0.18)	0.03
Absolute grip strength, kg ^d^	33.99 (33.36, 34.62)	33.58 (33.10, 34.07)	32.01 (30.97, 33.03)	0.01 (0.002, 0.01)	0.006
		Women			
Number of subjects	2167	2462	481		
Relative grip strength _max_ ^c^	0.93 (0.92, 0.94)	0.92 (0.91, 0.93)	0.90 (0.88, 0.92)	0.08 (0.02, 0.15)	0.02
Relative grip strength _mean_ ^c^	0.90 (0.89, 0.91)	0.89 (0.88, 0.90)	0.87 (0.85, 0.89)	0.08 (0.02, 0.15)	0.02
Relative grip strength _left_ ^c^	0.91 (0.90, 0.92)	0.90 (0.89, 0.91)	0.88 (0.85, 0.90)	0.08 (0.02, 0.15)	0.02
Relative grip strength _right_ ^c^	0.89 (0.88, 0.90)	0.88 (0.87, 0.89)	0.87 (0.84, 0.89)	0.08 (0.01, 0.15)	0.02
Absolute grip strength, kg ^d^	21.83 (21.59, 22.06)	21.65 (21.43, 21.86)	21.27 (20.78, 21.77)	0.003 (0.001, 0.006)	0.04

Results are shown as mean (95% confidence interval), except for numbers. Interaction effects between sex and handgrip strength on cognitive impairment were not significant (*p* from 0.27 to 0.87). Relative grip strength _max_: maximum of the average of the right or left grip strength divided by body mass index (BMI); Relative grip strength _mean_: mean of the average of both right and left grip strength divided by BMI; Relative grip strength _left_: average of the left grip strength divided by BMI; Relative grip strength _right_: the average of the right grip strength divided by BMI; Absolute grip strength: maximum of the average of right or left grip strength. ^a^: Adjusted for age, sex, education, smoking status, physical activity, fasting glucose, and systolic blood pressure; ^b^: Adjusted for age, sex, education, smoking status, physical activity, fasting glucose, systolic blood pressure, and BMI; ^c^: Adjusted for age, education, smoking status, physical activity, fasting glucose, and systolic blood pressure; ^d^: Adjusted for age, education, smoking status, physical activity, fasting glucose, systolic blood pressure, and BMI.

**Table 4 ijerph-19-06464-t004:** Multivariable linear regression of hand grip strength on Mini Mental State Examination (MMSE) groups * for males and females.

		MMSE Scores (0–30)	Adjusted Beta-Coefficient (95% CI)	*p* for Trend
	25–30	<25		
		Total		
Number of subjects	5996	662		
Relative grip strength _max_ ^a^	1.06 (1.05, 1.07)	1.04 (1.02, 1.07)	−0.02 (−0.04, 0.01)	0.12
Relative grip strength _mean_ ^a^	1.03 (1.02, 1.04)	1.01 (0.99, 1.03)	−0.02 (−0.05, 0.01)	0.11
Relative grip strength _left_ ^a^	1.04 (1.03, 1.05)	1.01 (0.99, 1.04)	−0.02 (−0.05, 0.01)	0.09
Relative grip strength _right_ ^a^	1.02 (1.01, 1.03)	1.01 (0.98, 1.03)	−0.02 (−0.04, 0.01)	0.21
Absolute grip strength, kg ^b^	24.83 (24.67, 24.98)	24.29 (23.80, 24.78)	0.02(0.004, 0.04)	0.06
		Men		
Number of subjects	1567	148		
Relative grip strength _max_ ^c^	1.45 (1.43, 1.47)	1.40 (1.34, 1.46)	−0.03 (−0.07, 0.006)	0.11
Relative grip strength _mean_ ^c^	1.41 (1.39, 1.43)	1.36 (1.30, 1.42)	−0.03 (−0.07, 0.007)	0.11
Relative grip strength _left_ ^c^	1.42 (1.40, 1.44)	1.37 (1.31, 1.43)	−0.03 (−0.07, 0.01)	0.09
Relative grip strength _right_ ^c^	1.40 (1.38, 1.41)	1.36 (1.29, 1.41)	−0.03 (−0.07, 0.01)	0.14
Absolute grip strength, kg ^d^	33.60 (33.22, 33.97)	32.39 (31.13, 33.65)	−0.002 (−0.003, 0.01)	0.11
		Women		
Number of subjects	4429	514		
Relative grip strength _max_ ^c^	0.93 (0.92, 0.93)	0.91 (0.89, 0.94)	−0.02 (−0.05, 0.02)	0.32
Relative grip strength _mean_ ^c^	0.90 (0.88, 0.90)	0.88 (0.86, 0.91)	−0.02 (−0.05, 0.02)	0.28
Relative grip strength _left_ ^c^	0.91 (0.89, 0.91)	0.89 (0.86, 0.91)	−0.02 (−0.06, 0.008)	0.15
Relative grip strength _right_ ^c^	0.89 (0.88, 0.90)	0.88 (0.86, 0.90)	−0.01 (−0.04, 0.02)	0.50
Absolute grip strength, kg ^d^	21.79 (21.64, 21.96)	21.36 (20.87, 21.85)	−0.001 (−0.003, 0.001)	0.10

*: Missing 148 subjects. Interaction effects between sex and handgrip strength on cognitive impairment were not significant (*p* from 0.07 to 0.58). Relative grip strength max: maximum of the average of the right or left grip strength divided by body mass index (BMI); Relative grip strength _mean_: mean of the average of both right and left grip strength divided by BMI; Relative grip strength _left_: average of the left grip strength divided by BMI; Relative grip strength _right_: the average of the right grip strength divided by BMI; Absolute grip strength: maximum of the average of right or left grip strength. ^a^: Adjusted for age, sex, education, smoking status, physical activity, fasting glucose, and systolic blood pressure; ^b^: Adjusted for age, sex, education, smoking status, physical activity, fasting glucose, systolic blood pressure, and BMI; ^c^: Adjusted for age, education, smoking status, physical activity, fasting glucose, and systolic blood pressure; ^d^: Adjusted for age, education, smoking status, physical activity, fasting glucose, systolic blood pressure, and BMI.

**Table 5 ijerph-19-06464-t005:** Multivariable linear regression of handgrip strength on Mini Mental State Examination (MMSE), all sub-components.

	Adjusted Beta-Coefficient (95% CI)	*p* Values
Relative grip strength _max_ ^a^		
Orientation	−0.07 (−0.14, 0.01)	0.91
Memory	0.01 (−0.004, 0.02)	0.23
Attention and calculation	0.01 (−0.10, 0.11)	0.90
Recall memory	0.07 (0.01, 0.15)	0.04
Language	0.03 (−0.03, 0.08)	0.37
Relative grip strength _mean_ ^a^		
Orientation	−0.07 (−0.14, 0.01)	0.09
Memory	0.008 (−0.004, 0.02)	0.21
Attention and calculation	0.008 (−0.10, 0.11)	0.88
Recall memory	0.08 (0.003, 0.15)	0.04
Language	0.03 (−0.03, 0.08)	0.37
Relative grip strength _left_ ^a^		
Orientation	−0.06 (−0.13, 0.02)	0.14
Memory	0.007 (−0.005, 0.02)	0.24
Attention and calculation	0.02 (−0.09, 0.12)	0.74
Recall memory	0.09 (0.02, 0.17)	0.01
Language	0.02 (−0.03, 0.08)	0.42
Relative grip strength _right_ ^a^		
Orientation	−0.07 (−0.15, 0.004)	0.06
Memory	0.008(−0.004, 0.02)	0.19
Attention and calculation	−0.002 (−0.116, 0.10)	0.98
Recall memory	0.06 (−0.02, 0.13)	0.14
Language	0.03 (−0.03, 0.08)	0.33
Absolute grip strength, kg ^b^		
Orientation	−0.002 (−0.006, 0.002)	0.26
Memory	−0.001 (−0.001, 0.001)	0.09
Attention and calculation	0.001 (−0.004, 0.006)	0.61
Recall memory	0.005 (0.002, 0.009)	0.003
Language	0.001 (−0.002, 0.004)	0.44

Relative grip strength _max_: maximum of the average of the right or left grip strength divided by body mass index (BMI); Relative grip strength _mean_: mean of the average of both right and left grip strength divided by BMI; Relative grip strength _left_: average of the left grip strength divided by BMI; Relative grip strength _right_: the average of the right grip strength divided by BMI; Absolute grip strength: maximum of the average of right or left grip strength. ^a^: Adjusted for age, sex, education, smoking status, physical activity, fasting glucose and systolic blood pressure; ^b^: Adjusted for age, sex, education, smoking status, physical activity, fasting glucose, systolic blood pressure and BMI.

## Data Availability

All data generated or analyzed during this study are included in this article. Further enquiries can be directed to the corresponding author.

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
