# Peer review of "Association of Hand Grip Strength with Mild Cognitive Impairment in Middle-Aged and Older People in Guangzhou Biobank Cohort Study"

_ijerph, 2022, doi:10.3390/ijerph19116464_

Round 1

Reviewer 1 Report

This was an interesting piece of work using a large dataset of adults >6000 adults aged from 50 years. There is some originality in the findings that are worth publishing.

General comments

P values should be p-values. In all tables and text relating to p-values, zero is not required
e.g. p < .001 , not p< 0.001.

For the discussion consider the following. What are the conceptual connections between the Delayed Word Recall Test and aspects of the MMSE. Did the DWRT emerge from a modified version of the MMSE? See Lyness, S. A., Lee, A. Y., Zarow, C., Teng, E. L., & Chui, H. C. (2014). 10-minute delayed recall from the modified mini-mental state test predicts Alzheimer's disease pathology. Journal of Alzheimer's Disease39(3), 575-582.

Section comments

Abstract
Line 23 MMSE scores

Line  22-23. No significant association between handgrip strength and total MMSE scores was found in the total sample or by sex (P from 0.06 to 0.50). p-values do not require 0.

What are the issues with MMSE (literature?)

Introduction

Line 33 ‘characterizing’ to ‘characterized’

Handgrip strength is confounded by body size and authors have used relative handgrip strength as well to adjust for BMI as a proxy for body size (handgrip / BMI). Could BMI be included as a covariate or height and weight could be included as covariates to compare findings with those reported? Are the findings consistent with those reported in this paper?

Line 53 – Remove ‘et al.’ and replace with ‘etc.’

Line 82 You say the procedure was repeated three times. Were different words used each time? This may be worth noting in the discussion. When you mention the ‘fourth’ repetition, should this not be the ‘third’ ?

Line 83 - Change ‘min’ to ‘minutes’.

Mean scores in brackets should be consistently changed to (M=?, SD=).

Line 87 – exchange  ‘to’ with  ‘into’.

The cut-off scores are not exactly one standard deviation below the means for DWRT or MMSE but are rounded to the nearest whole number. This is worth stating.

Are there references to support the values of the cut-off scores?

Lines 97-98 state that an average grip strength score was calculated but then it states the maximal score was used. Were there two variables here (‘maximal’ grip strength and ‘average’)? Were both used in the analysis? Provide clarification of the term ‘maximal’. The authors then use the term maximum relative grip strength. The interchangeable use of ‘maximal’ and ‘maximum’ here is confusing.

Line 106 – change ‘was’ to ‘were’.

Line 108 – make ‘number’ and ‘percentage’  and line 113 ‘coefficient’ and ‘confidence interval’  plural.

Line 113 states “potential confounders….”. This is repeated on the previous line.

Line 116 - Use small p when describing probability values throughout the paper.

Line 117 change ‘of’ to ‘in’.

Line 118 - change ‘sex interaction’ to ‘a sex interaction’.

Line 119 consider removing or rephrasing “and all analyses were done on total participants and following sex stratification” as this is unclear.

P124 change ‘of them’ to ‘of these’

Line 126 – states “…compared to the cognitively non-impaired group, cognitive impairment was greater in those with older age, lower education, active physical activity(?) and higher systolic blood pressure…”. I would refer to DWRT scores specifically here as MMSE also measure cognitive impairment but is not stated in table 1. Also , the word activity should be inactivity.

Line 128 remove ‘more’

Line 134 – change MMSE scores and it’s domains to “MMSE overall scores and all the sub-domains”

Lines 113-118 - What is the rationale for table 2 results ? If we want to know whether MMSE scores are correlated with DWRT scores then why not use Pearson correlations (or partial correlations to adjust for the covariates stated in the regression).

If you want to use the regression table, state if the regression coefficients were standardized or unstandardized and state this also in the table 2 title. R2 values would also be informative here as well.

Table 3 title needs refined to also include mention of “Multiple regression of grip strength indices on MMSE total scores and all sub-components for males and females separately”.

Line 147 – change to ‘linear trends were…’

Line 175 – Should this be indented or is this the textual account of results?

Table 5 - Are the results in table 5 (whole sample) consistent for both males and females separately? This is also worth a comment in the discussion.

Line 200 – ‘across-sectional’ change to ‘a cross-sectional’

Line 231- add space to  ‘52acutely’

Reviewer 2 Report

See attachment.

Reviewer 3 Report

This is an interesting and very current topic, given that nowadays dementia is very worrying in people from different countries and appears in people who are not very old.

Although we are in the presence of a study with a very large sample, I believe that the statistical approach needs to be improved, as it has some inaccuracies or omissions.

I leave several comments trying to clarify the text of the article and the statistical approaches used and others that seek to correct approaches I understand to be incorrect.

  • In line 21 "in all participants" must be deleted.
  • In line 21 do the authors mean by the expression "no sex interaction (p=0.58) was found..."? It is the interaction of sex with which variables?
  • In line 49, there are a lot of articles that are not indexed on Pubmed, so I find it very reductive that the authors did just a search on this platform.
  • Don’t the authors have more recent data? In 15 years, there has been a lot of evolution in this field (for example, the appearance of more cases of dementia than usual and in younger people) so the results may not be very adequate in the present.
  • It is unclear why the authors divided the participants into 3 groups according to the DWRT value obtained. This is an ordinal variable that could be analyzed as such. With three groups you lost information and I can’t explain how you did it. On the other hand, it is not advisable to calculate means and standard deviations for data on ordinal scales, as they may not be representative of the distribution (in this case there are only 11 different values and we do not know if the distribution is, for example, very skewed). And what is the scientific basis for using a unit of standard deviation for the cutoff point below and above the mean?
  • Similarly, what is the scientific basis for using a standard deviation unit below and above the mean value of the MMSE for the cut-off point? Is having less than 25 (out of a maximum of 30) poor cognitive? Why? Are there any studies that support what they did? What is the distribution of MMSE values in this sample? With two groups, you lost information.
  • Why did they take the average of the two-grip strength of each hand and not consider the individual values? And why two measurements?
  • Have you tested the assumptions of the approaches ANOVA, ANCOVA and multilinear regression models? How did you do it? And what were the results? Did it always meet the assumptions?
  • The authors summarized all continuous variables with mean and standard deviation, but in skewed or non-mesokurtic distributions these values are not the most adequate. Have you analyzed the distribution of each continuous variable?
  • Please review the text of the last two periods of point 2.4.
  • Lines 136 to 138 repeat what is in table 2. On the other hand, in line 135 they mention that for memory the value p=0.04 does not allow concluding that the variable is significant. What is the significance level used in the study?
  • In Tables 2, 3 and 4, the adjusted beta coefficients result from which statistical approach?
  • Are you modelling memory (4 different values), attention and calculation (5 different values) and recall memory (4 different values) as if they were continuous? And calculate the mean and standard deviation for these variables? This doesn't make any sense!!
  • Simplify the text below tables 3, 4 and 5 as it is unnecessarily repetitive.

Round 2

Reviewer 3 Report

The authors improved the article and answered several questions asked. However, there are still some questions that urgently need to be answered, as they are of a methodological nature:

1) MMSE values theoretically range between 0 and 30. How does a mean of 27.7 correspond to asymmetrical distribution? Please indicate the minimum, maximum, mean and quartile values for the MMSE.

2) I still don't understand why you didn't model the MMSE as a continuous variable. The loss of information is clear in this case. Furthermore, a 2004 reference to justify a cut-off point is very poor.

3) Grip strength of each hand was tested two times, and the average value of each hand was calculated. You still didn't answer why using the mean and not the two individual values.

4) You used ANOVA, ANCOVA and multilinear regression models. The text remains unclear in which situations they use each approach. On the other hand, the assumptions of these approaches are not just the normality and homogeneity of variances. Consequently, they must complete the answer to the question I posed in the previous review.

5) In response to the first review, you responded that the adjusted beta coefficients were from multivariable generalized linear regression. What do you mean by generalized? And, on the other hand, the memory and recall memory variables were modelled as categorical or as continuous variables? In particular, memory appears to have nearly constant values, as the mean is almost equal to the maximum.
